# Continuous Pseudo-Labeling from the Start

**Dan Berrebbi***

Carnegie Mellon University
dberrebb@andrew.cmu.edu

**Ronan Collobert, Samy Bengio,**
**Navdeep Jaitly, Tatiana Likhomanenko**

Apple
{collobert,bengio,njaitly,antares}@apple.com

## Abstract

Self-training (ST), or pseudo-labeling has sparked significant interest in the automatic speech recognition (ASR) community recently because of its success in harnessing unlabeled data. Unlike prior semi-supervised learning approaches that relied on iteratively regenerating pseudo-labels (PLs) from a trained model and using them to train a new model, recent state-of-the-art methods perform 'continuous training' where PLs are generated using a very recent version of the model being trained. Nevertheless, these approaches still rely on bootstrapping the ST using an initial supervised learning phase where the model is trained on labeled data alone. We believe this has the potential for over-fitting to the labeled dataset in low resource settings and that ST from the start of training should reduce over-fitting. In this paper we show how we can do this by dynamically controlling the evolution of PLs during the training process in ASR. To the best of our knowledge, this is the first study that shows the feasibility of generating PLs from the very start of the training. We are able to achieve this using two techniques that avoid instabilities which lead to degenerate models that do not generalize. Firstly, we control the evolution of PLs through a curriculum that uses the online changes in PLs to control the membership of the cache of PLs and improve generalization. Secondly, we find that by sampling transcriptions from the predictive distribution, rather than only using the best transcription, we can stabilize training further. With these techniques, our ST models match prior works without an external language model.

## 1 Introduction

The past few years have witnessed a growth in methods that leverage large amount of unlabeled data in domains such as speech, vision and language to produce state-of-the-art results, e.g. Baevski et al. (2020; 2022); Chen et al. (2020a); Caron et al. (2021); He et al. (2022); Cai et al. (2022); Brown et al. (2020); Ramesh et al. (2021). Amongst the techniques that have made this possible are self-supervised learning (SSL) and self-training (ST) (Scudder, 1965; Lee, 2013). While SSL is typically used in unsupervised settings, ST is applied in supervised settings where labeled data can be extended with unlabeled data that is labeled using a prior model, a process known as pseudo-labeling (PL). These techniques can reduce the burden of expensive labeling processes while successfully train data hungry models such as transformers using large quantities of unlabeled data.

Current state-of-the-art SSL methods in speech (Baevski et al., 2020; Hsu et al., 2021; Baevski et al., 2022; Chung et al., 2021) are typically trained in two phases. First, the models are pre-trained on thousands of hours of unlabeled speech, and then they are further adapted by fine-tuning on the actual task of automatic speech recognition (ASR) using a smaller supervised set. However, because the pre-training (PT) phase is task agnostic, self-supervision can under-perform on a specific downstream task (Talnikar et al., 2021; Dery et al., 2022). Further, SSL pre-training leads to a more complicated pipeline involving multiple phases. By contrast, ST algorithms also use unlabeled data but do not require phases of training with different objectives that makes the training pipeline simpler.

In this paper, we focus on recent ST algorithms that perform 'continuous training' of a single model. In contrast to earlier ST training methods that iterate between generating PLs over the entire unlabeled dataset and training a model (teacher-student) (Synnaeve et al., 2020; Kahn et al., 2020a; Zhang

---

*Work done during internship at Apple.

Table 1: Continuous ST (using slimIPL) with different pre-training steps ($M$) using a 10h dataset reveals that more pre-training can lead to worse results (we show word error rate, WER, on dev-clean).

| $M$ | 10k | 20k | 40k |
|-----|------|------|------|
| WER | 14.3 | 17.1 | 22.9 |

et al., 2020; Park et al., 2020), here pseudo-labels (PLs) are generated online with a very recent version of the model (Xu et al., 2020; Likhomanenko et al., 2021a; Manohar et al., 2021; Higuchi et al., 2021; 2022a;b) and training is faster and more resource-efficient. One of the main challenges for continuous ST is training stability (Likhomanenko et al., 2021a; Higuchi et al., 2021; 2022b; Cai et al., 2022). While these prior works use various techniques for stabilization, one common ingredient is that models are initially trained on labeled data for $M$ steps. slimIPL (Likhomanenko et al., 2021a) showed robustness to $M$ in some settings, but a well-established recipe does not seem to exist for the case of small labeled datasets (aka. the low resource setting). Indeed, we find that more pre-training steps, compared to what was shown previously in Likhomanenko et al. (2021a), can lead to worse results (see Table 1). We hypothesize that this is due to over-fitting to the labeled set early in training in low resource settings and in this paper we try to improve results by doing ST without any pre-training (i.e. $M = 0$). However, in our experiments, off-the-shelf slimIPL diverges early in training in low resource settings, so we developed methods to address this problem which we summarize here:

- We show that sampling transcriptions from the output distribution instead of using the best transcription makes ST robust and stable, especially when no pre-training is performed.
- We propose a new curriculum for controlling the PL distribution during training. The curriculum uses the Levenshtein distance between PLs at different time steps to control how PLs are updated, and how unsupervised examples are chosen for training.

For the first time, with these strategies we show that continuous PL can be done from the very start of the training matching prior works without an external language model.

## 2    EXPERIMENTAL SETUP AND RELATED METHODS

**Data**    All our experiments are performed using the LibriSpeech dataset (Panayotov et al., 2015). We use the *train-clean-360* and *train-other-500* regular subsets as unlabeled data, and consider either a subset of 10h randomly drawn from *train-clean-100*, or the full 100h set (*train-clean-100*) as labeled data. Comparisons with existing works are also provided using the 10h subset from Libri-Light (Kahn et al., 2020b)[1]. In addition, we evaluate the final configuration of our methods on the Common Voice dataset Ardila et al. (2020) for French language where we sample 10h and 100h from the train set to use as labeled data and the rest as unlabeled data (see Appendix A.3).

**Acoustic model**    Following Likhomanenko et al. (2021a), models are trained with English letters token set[2], the Connectionist Temporal Classification Graves et al. (2006) (CTC) loss, identical SpecAugment (Park et al., 2019) parameters, and Adagrad optimizer (Duchi et al., 2011). The acoustic model is the same transformer architecture that was introduced in slimIPL, except that we encode positions with either absolute sinusoidal positional embedding (Vaswani et al., 2017) or the recently proposed CAPE (Likhomanenko et al., 2021b) instead of relative positional embedding (Shaw et al., 2018). This allows us to speed up training (by 2-3x) and decrease the memory footprint significantly. All models are trained on 8 GPUs for a maximum of 500k updates. We use either a static batch of 8 examples or a dynamic batch that packs $\sim$ 290s of audio per GPU.

**Continuous pseudo-labeling (PL) in ASR**    Let $L = \{\boldsymbol{x}_i, \boldsymbol{y}_i\}$ and $U = \{\boldsymbol{x}_j\}$ be the labeled and unlabeled datasets, respectively. We consider a semi-supervised PL approach where an acoustic model

---

[1]Libri-Light 10h subset contains only 24 speakers drawn from the *whole* LibriSpeech (from both clean and noisy subsets). To keep our experiments consistent, and also to assess domain transfer to the unlabeled noisy subsets, we reconstructed the 10h set from the *train-clean-100*, sampling randomly from the speakers and retaining the original 250 speakers from this subset.

[2]26 letters augmented with the apostrophe and a word boundary token.

---

**Algorithm 1:** slimIPL algorithm and our proposed changes (red → deletion and green → addition)

---

**Inputs:** labeled $L = \{\boldsymbol{x}_i, \boldsymbol{y}_i\}$ and unlabeled $U = \{\boldsymbol{x}_j\}$ data, $\tilde{\boldsymbol{x}} = augmentation(\boldsymbol{x})$, initialization $\boldsymbol{\theta}^0$,
    cache $\mathcal{C} = \{\}$, learning rate $\eta_k$, losses $\mathcal{L}_L$ and $\mathcal{L}_U$, parameters $M, N_L, N_U, p_{out}$ and $C$
    PL function $PL(\boldsymbol{x}; \boldsymbol{\theta}, \tau) = PL(\boldsymbol{x}; \boldsymbol{\theta})$ defined via Eq. (2)
    PL function $PL(\boldsymbol{x}; \boldsymbol{\theta}, \tau)$ defined via sampling with temperature $\tau$ (see Section 4.2)
**Result:** Acoustic model $\mathcal{A}(\boldsymbol{x}; \boldsymbol{\theta})$

1  // Initial pre-training (PT) phase : train only on labeled samples
2  Train $\mathcal{A}$ on $(\boldsymbol{x}, \boldsymbol{y}) \in L$ for $M$ steps:
3     $\boldsymbol{\theta}^{k+1} = \boldsymbol{\theta}^k - \eta_k \nabla \mathcal{L}_L(\mathcal{A}(\tilde{\boldsymbol{x}}; \boldsymbol{\theta}^k), \boldsymbol{y}), k = \overline{1, M}$
4  Decrease model's $\mathcal{A}(\boldsymbol{x}; \boldsymbol{\theta})$ dropout
5  // Train on labeled data while filling the cache
6  **for** $k = \overline{M+1, M+C}$ **do**
7     For random $\boldsymbol{x} \in U$ generate $\hat{\boldsymbol{y}} = PL(\mathcal{A}_{inference}(\boldsymbol{x}; \boldsymbol{\theta}^k), \tau)$ and $\mathcal{C} \leftarrow \mathcal{C} \bigcup \{(\boldsymbol{x}, \hat{\boldsymbol{y}})\}$
8     $\boldsymbol{\theta}^{k+1} = \boldsymbol{\theta}^k - \eta_k \nabla \mathcal{L}_L(\mathcal{A}(\tilde{\boldsymbol{x}}; \boldsymbol{\theta}^k), \boldsymbol{y}), (\boldsymbol{x}, \boldsymbol{y}) \in L$
9     $\tau = \max(0, 1 - k/K)$
10  // Continuous pseudo-labeling training with the cache
11  **repeat**
12     **if** $rand(0, 1) < N_L/(N_L + N_U)$ **then**
13         Draw $(\boldsymbol{x}, \boldsymbol{y}) \in L$ and $\boldsymbol{\theta}^{k+1} = \boldsymbol{\theta}^k - \eta_k \nabla \mathcal{L}_L(\mathcal{A}(\tilde{\boldsymbol{x}}; \boldsymbol{\theta}^k), \boldsymbol{y})$
14     **else**
15         Draw $b = (\boldsymbol{x}, \boldsymbol{y}) \in \mathcal{C}$ and $\boldsymbol{\theta}^{k+1} = \boldsymbol{\theta}^k - \eta_k \nabla \mathcal{L}_U(\mathcal{A}(\tilde{\boldsymbol{x}}; \boldsymbol{\theta}^k), \boldsymbol{y})$
16         $\hat{\boldsymbol{y}} = PL(\mathcal{A}_{inference}(\boldsymbol{x}; \boldsymbol{\theta}^k, \tau))$ // Compute current model state PL
17         $p_{out} = TER(\boldsymbol{y}, \hat{\boldsymbol{y}})$ if $k < K$ else $p_{out} = 1$ // Compute dynamic $p_{out}$
18         **if** $rand(0, 1) < p_{out}$ **then**
19             For random $\boldsymbol{x}' \in U$ generate $\hat{\boldsymbol{y}}' = PL(\mathcal{A}_{inference}(\boldsymbol{x}'; \boldsymbol{\theta}^k, \tau))$ and $\mathcal{C} \leftarrow \mathcal{C} \setminus b \bigcup \{(\boldsymbol{x}', \hat{\boldsymbol{y}}')\}$
20         **else**
21             $\mathcal{C} \leftarrow \mathcal{C} \setminus b \bigcup \{(\boldsymbol{x}, \boldsymbol{y})\}$ // Same sample and PLs back into the cache
22             $\mathcal{C} \leftarrow \mathcal{C} \setminus b \bigcup \{(\boldsymbol{x}, \hat{\boldsymbol{y}})\}$ // Same sample but new PLs back into the cache
23     $k \leftarrow k + 1$
24     $\tau = \max(0, 1 - k/K)$
25  **until** convergence or maximum iterations are reached

---

$\mathcal{A}(\boldsymbol{x}; \boldsymbol{\theta})$ with model parameters $\boldsymbol{\theta}$ is continuously trained on a combination of $L$ and a pseudo-labelled set derived from $U$. The model is trained by minimizing a loss

$$\mathcal{L}(\boldsymbol{\theta}) = \mathcal{L}_L(\boldsymbol{\theta}) + \lambda \mathcal{L}_U(\boldsymbol{\theta}), \tag{1}$$

where $\lambda \in \mathbb{R}^+$ is a tunable hyper-parameter controlling the importance of unlabeled data. The loss for labeled data is defined as $\mathcal{L}_L(\boldsymbol{\theta}) = -\mathbb{E}_{(\boldsymbol{x}, \boldsymbol{y}) \sim L} \log p_{\boldsymbol{\theta}}(\boldsymbol{y}|\boldsymbol{x})$, where $p_{\boldsymbol{\theta}}(\boldsymbol{y}|\boldsymbol{x})$ is the conditional distribution defined by $\mathcal{A}(\boldsymbol{x}; \boldsymbol{\theta})$. The loss for unlabeled data is defined as $\mathcal{L}_U(\boldsymbol{\theta}) = -\mathbb{E}_{\boldsymbol{x} \sim U} \log p_{\boldsymbol{\theta}}(\hat{\boldsymbol{y}}|\boldsymbol{x})$, where $\hat{\boldsymbol{y}}$ is the PL transcription for a data point generated using the model being trained. Specifically,

$$\hat{\boldsymbol{y}} = \operatorname*{argmax}_{\boldsymbol{y}} \log p_{\boldsymbol{\theta}}(\boldsymbol{y}|\boldsymbol{x}). \tag{2}$$

Continuous PL keeps updating the pseudo-labels via Eq. (2), as the model trains. This procedure is prone to divergence, as without any constraint PLs can self-reinforce rapidly to a trivial distribution.

**Methods to stabilize training**   Several approaches have been proposed to stabilize continuous PL. A pre-training phase (PT) on the supervised data only (optimizing the loss $\mathcal{L}_L(\boldsymbol{\theta})$ for $M$ updates) is always a key component. For e.g. in Chen et al. (2020b) PT is performed until full convergence. Another technique is the use of an exponential moving average (EMA) of the acoustic model to generate the pseudo-labels in Eq. (2) (Likhomanenko et al., 2021a; Manohar et al., 2021; Higuchi et al., 2021; 2022b; Zhang et al., 2022).

**slimIPL**   To avoid the significant memory footprint of EMA Likhomanenko et al. (2021a) introduced slimIPL, which uses a dynamic cache instead of the EMA to stabilize the training. The cache maintains a set of unlabeled samples $U^{\mathcal{C}}$ (with fixed size $|U^{\mathcal{C}}| = C$) and their associated PLs, generated by previous model states. After the pre-training phase, slimIPL minimizes the loss in Eq. (1), using the

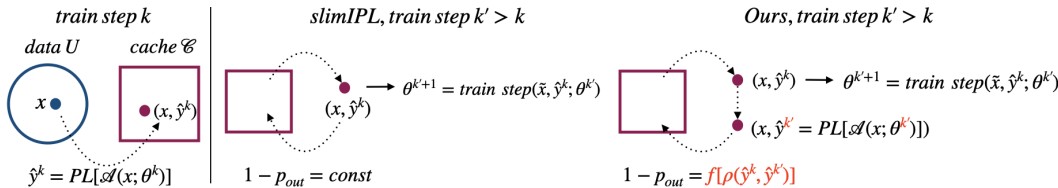

Figure 1: Comparison between slimIPL (left) and how we control the cache by using PL evolution (right). The constant $p_{out}$ from slimIPL now is dynamic and computed based on the PL evolution.

unlabeled subset $U^{\mathcal{C}}$, which is itself updated as training goes: at each iteration, slimIPL removes a sample from the cache with probability $p_{out}$, replacing it with a new one $x \in U$ along with its generated PL. More details about slimIPL can be found in Algorithm 1 and in Figure 1.

**PLs selection**  Pseudo-labels selection can help to achieve better convergence by filtering out noisy PLs that prevent model from faster training. There are also a lot of efforts on the curriculum pseudo-labeled data selection: e.g. confidence filtering (Zhang et al., 2021) or assigning weights to pseudo-labeled data based on the model uncertainty estimation (Huang et al., 2022). One of the recent works (Zhang et al., 2022) in ASR proposes to use PLs curriculum filtering based on the Levenshtein distance between PLs generated for original and weakly augmented inputs. Later we will see that our idea is based solely on the PL evolution rather than on input augmentation.

**Relation to consistency regularization**  Popular consistency regularization methods (Sajjadi et al., 2016; Laine & Aila, 2016; Sohn et al., 2020; Berthelot et al., 2019) leverage the idea that a model should output the same distribution for an unlabeled example even after it has been augmented. In this paper we take inspiration from these works but we focus on an orthogonal view: we consider distances between model outputs at different time steps. Also, contrary to consistency regularization, we do not use this distance as an objective function to train a model but as a data selection criterion.

**Hyper-parameter selection**  All hyper-parameters and model selections are performed using *dev-clean* and *dev-other* sets. We report final token (TER) or word (WER) error rates on *test-clean* and *test-other* sets. In all experiments, we only tune $(C, p_{out}, M, \lambda)$ from the training procedure while everything else is kept as in the slimIPL paper. By default we use $C = 1000$, $\lambda = 1$, $M = 0$. In most experiments we try 3 different random seeds and report metric mean and standard deviation.

## 3  MOTIVATION

Existing continuous PL approaches rely on a two-step process: first pre-training (PT) on labeled data only, then continue the model training with both labeled and unlabeled data. While PT is known to be critical for the stability of continuous PL, we are interested in this work to find ways to remove the PT phase to simplify the whole procedure, and possibly improve the overall performance, both in terms of convergence speed and final WER.

**PT improves the final WER**  Initial experiments with slimIPL, Table 2, show that with even its simple cache strategy used to stabilize training, PT helps improving the final WER. It is not surprising, as without PT, PLs are of poor quality ($> 90\%$ WER) at the beginning of training as the model mostly produces random outputs. Careful tuning of the number of PT steps is however important, especially in low-resource supervised settings, as shown in Table 1.

**Caching as a replacement for PT**  Vanilla continuous PL is very similar to slimIPL with $p_{out} = 1$ (see Section 2). With the caching strategy, slimIPL picks unlabeled samples (and their associated PLs) from a cache when needed, and immediately replaces these examples with new unlabeled samples (and their new PLs). This allows to always use PLs generated from a previous version of the trained model, while efficiently computing these PLs. While being simple, we observe in Table 2 that this approach is enough to stabilize continuous PL, assuming a large enough cache.

Table 2: Continuous PL w/ and w/o pre-training (PT) phase for slimIPL. 'DV' states for divergence.

| Data | $p_{out}$ | dev-clean WER | | dev-other WER | |
|------|-----------|---------------|-------|---------------|-------|
| | | w/o PT | w/ PT | w/o PT | w/ PT |
| 10h | 1 | $23.3_{1.7}$ | 13.8 | $32.1_{1.3}$ | 17.5 |
| 10h | 0.1 | DV | 11.4 | DV | 14.0 |
| 100h | 1 | $4.5_{0.1}$ | 3.1 | $10.6_{0.3}$ | 8.1 |
| 100h | 0.1 | DV | 3.6 | DV | 7.5 |

**When to update the PLs from the cached samples is critical**   In slimIPL (Algorithm 1), each sample $(\boldsymbol{x}, \hat{\boldsymbol{y}})$ in the cache $\mathcal{C}$ at step $k'$ has a PL $\hat{\boldsymbol{y}} = PL(\mathcal{A}(\boldsymbol{x}; \boldsymbol{\theta}^k))$ that was generated with the model $\boldsymbol{\theta}^k$ at step $k < k'$ when it was added to the cache. After using the sample $(\boldsymbol{x}, \hat{\boldsymbol{y}})$ for training, slimIPL adds it back into the cache with probability $1 - p_{out}$, leaving its corresponding PLs *unchanged*. We found however that updating PLs with the current model state $\hat{\boldsymbol{y}} = PL(\mathcal{A}(\boldsymbol{x}; \boldsymbol{\theta}^{k'}))$ improves final WER performance. See Table 3, which compares the original slimIPL strategy ('old'), with the one where the PLs are updated when a sample has been selected in the cache ('new'). For that reason, in the following experiments, we will be using $\hat{\boldsymbol{y}} = PL(\mathcal{A}(\boldsymbol{x}; \boldsymbol{\theta}^{k'}))$ as a PL strategy, when keeping a sample back into the cache.

**Controlling cache contents dynamically can improve WER**   When the cache is updated less often ($p_{out} < 1$), we see in Table 2 that one may improve the WER, but then PT is essential to avoid any divergence. In Likhomanenko et al. (2021a), the authors of slimIPL have reported robustness (in terms of test WER) with respect to $p_{out}$. However, our experiments reported in Table 3 and Figure 3b in Appendix C reveal different learning dynamics for different values of $p_{out}$: our ablations with specific schedules on the probability $p_{out}$ suggest that models without a PT phase would benefit more from low $p_{out}$ at the beginning of training, which would make training easier initially by letting the model focus on the same examples. In addition, later in training, the training procedure might benefit from high $p_{out}$, as seeing a wider range of examples may lead to more stability. While we observe significant changes in dynamics with 10h of supervision, with larger labeled set (100h) the different strategies do not make such a huge difference.

The above observations suggest that by dynamically controlling how the cache evolves we can improve results in limited data settings. One possible way of doing this is by using a strategy that depends on the rate of evolution of PLs in the cache. In the next section we present such a method.

Table 3: Strategies of PLs and cache renewing (w/o PT phase). When $p_{out} < 1$ and sample goes back into the cache, we compare models using the same PL as it was $\hat{\boldsymbol{y}} = PL(\mathcal{A}(\boldsymbol{x}; \boldsymbol{\theta}^k))$ (old) or the newly re-generated PL $\hat{\boldsymbol{y}} = PL(\mathcal{A}(\boldsymbol{x}; \boldsymbol{\theta}^{k'}))$ (new). For cache renewing, we compare static $p_{out}$ and simple scheduling with $p_{out}$ being different before and after $130k$ steps.

| $p_{out}$ | PLs | 10h, WER | | 100h, WER | |
|-----------|-----|-----------|-----------|-----------|-----------|
| | | dev-clean | dev-other | dev-clean | dev-other |
| 1 | - | $23.3_{1.7}$ | $32.1_{1.3}$ | $4.5_{0.1}$ | $10.6_{0.3}$ |
| 0.1 | old | DV | DV | DV | DV |
| 0.1 | new | $15.3_{0.6}$ | $25.4_{0.4}$ | $4.5_{0.1}$ | $10.4_{0.1}$ |
| $1 \rightarrow 0.1$ | old | $23.0_{1.1}$ | $32.1_{0.4}$ | $4.5_{0.1}$ | $11.0_{0.0}$ |
| $1 \rightarrow 0.1$ | new | $24.8_{1.4}$ | $36.1_{0.5}$ | $\mathbf{4.4}_{0.0}$ | $\mathbf{10.2}_{0.1}$ |
| $0.1 \rightarrow 1$ | old | DV | DV | DV | DV |
| $0.1 \rightarrow 1$ | new | $\mathbf{13.7}_{0.8}$ | $\mathbf{20.7}_{0.8}$ | $4.8_{0.1}$ | $11.3_{0.1}$ |

## 4   METHODS OF STABLE TRAINING

### 4.1   CONTROLLING CACHE BY USING PL EVOLUTION

Let's consider an example $\boldsymbol{x} \in U$ to be put into the cache at training step $k$, see Figure 1. Its PL is defined as $\hat{\boldsymbol{y}} = PL(\mathcal{A}(\boldsymbol{x}; \boldsymbol{\theta}^k)) = PL(\boldsymbol{x}; k)$. At step $k' > k$, this example $(\boldsymbol{x}, \hat{\boldsymbol{y}})$ is selected

from the cache and the model is updated to $\boldsymbol{\theta}^{k'+1}$ using the gradient of the loss. Unlike slimIPL, the probability of removing the example from the cache is not constant anymore. Instead, $p_{out}$ is dynamically computed at step $k'$ **for sample $\boldsymbol{x}$** that is selected from the cache as follows:

$$p_{out}(\boldsymbol{x}; k) = f[\rho(PL(\boldsymbol{x}; k), PL(\boldsymbol{x}; k'))] \tag{3}$$

where $\rho$ is the Levenshtein edit-distance, and $f$ the function that encapsulates how evolution in PLs should determine the rate at which examples are removed from the cache. Using different choices of $f$ we can consider different ways of actively controlling the cache (and hence the model training) using the evolution of the PLs. We consider simple functions $f : x \mapsto x$ and $f : x \mapsto 1 - x$. The first function encourages the cache to maintain examples whose PLs are stable, which might lead to slower learning. The second function maintains examples whose PLs are changing fast which might lead to faster learning but less stable behavior.

Note that while we explained the method using a single example $\boldsymbol{x}$ from the unlabelled set, in practice we operate the algorithm on a batch level, and the statistics are computed over a full batch of examples, which are all put back in the cache or removed together.

## 4.2 Alignment Sampling

As discussed in Section 3 training instability shows up as the acoustic model distribution $\mathcal{A}(\boldsymbol{x}; \boldsymbol{\theta}^k)$ collapses to a degenerate distribution, e.g. empty transcriptions. While a cache and/or an exponential moving average model can stabilize training, they do not resolve the issue entirely, especially in the low data regime, with no pre-training, and the model often collapses to a degenerate solution. Even our proposed method above (see Section 4.1) is susceptible to this collapse on the 10h dataset.

In order to overcome the collapse issue and still make use of unlabeled data as early as possible, we propose to sample targets from the token distribution for every frame (Likhomanenko et al., 2022). We believe that sampling PLs around the most probable hard labels is an effective stabilization technique which works by adding appropriate noise to the targets: it is a way to enforce a lower bound on the entropy of the label distribution which mitigates the collapse issue[3]. As the model is learnt with CTC, every per frame predicted distribution $p_{\boldsymbol{\theta}}^t(w|\boldsymbol{x}), w \in \boldsymbol{w}$ for token set $\boldsymbol{w}$ and time frame $t$ is considered to be independent. Thus, for every audio frame, we sample a token label $w_t \sim p_{\boldsymbol{\theta}}^t(w|\boldsymbol{x})$. A temperature $\tau$ is introduced to smooth the distribution obtained from the model. After the frame level labels are sampled, they are transformed into the transcription by deduplicating consecutive repetitions of the same output token, and removing the left over auxiliary blank tokens[4].

**Sampling Temperature Schedule**   As $\tau \to \infty$ the distribution over tokens $p_{\boldsymbol{\theta}}^t(w|\boldsymbol{x}, \tau)$ approaches the uniform one, the PL sequence of tokens becomes purely random. On the other hand, as $\tau \to 0$ the distribution approaches the argmax function which is equivalent to the hard labels in slimIPL. We find that $\tau > 1$ performs poorly. With $\tau = 1$ a model avoids divergence at the beginning of training but end up with worse final performance than hard PLs ($\tau = 0$): this happens mostly because of larger noise presence due to sampling (quality of PLs is observed being worse). Lower temperatures, e.g. $\tau = 0.1$, give indistinguishable results from hard PLs ($\tau = 0$). These observations suggest that decreasing temperature as training proceeds can stabilize training at the beginning and benefit from less noisy PLs in the end. We found that simple linear schedule for $\tau$ from 1 to 0.1 works well.

The summary of our proposed methods on top of slimIPL is given in Algorithm 1.

## 5 Results

### 5.1 Dynamic Selection for Pseudo-Labeled Samples

In Table 4 we show results from using only the method introduced in Section 4.1. We experiment with token error rate (TER) distance computed between PLs on an entire batch and the two functions as discussed above. For both settings of 100h and 10h of supervised data the proposed dynamic

---

[3]With no regularization (cache, and/or alignment sampling), the PL procedure often collapses to generating just blanks very quickly (Likhomanenko et al., 2021a) – it is biased, has 100% WER, but has no variance. Alignment sampling avoids this by generating noisy targets that have variance.

[4]E.g. alignment 'cc###aatttt#' will be transformed into 'cat', where # is a CTC blank token.

Table 4: WER on *dev-clean* and *dev-other* for different cache selection methods ($p$). We use either $p_{out} = p$ or a strategy where $p_{out} = p$ for the first 130K steps, switching to $p_{out} = 1$ afterwards, as shown in Section 3. Alignment sampling from Section 4.2 is not used.

| | 10h | | | | 100h | | | |
| --- | --- | --- | --- | --- | --- | --- | --- | --- |
| | $p_{out} = p$ | | $p_{out} : p \to 1$ | | $p_{out} = p$ | | $p_{out} : p \to 1$ | |
| $p$ | clean | other | clean | other | clean | other | clean | other |
| 0.1 | $15.3_{0.6}$ | $25.4_{0.4}$ | $13.7_{0.8}$ | $20.7_{0.8}$ | $4.5_{0.1}$ | $10.6_{0.3}$ | $4.8_{0.1}$ | $11.3_{0.1}$ |
| $TER[PL(k), PL(k')]$ | $\mathbf{14.7}_{0.5}$ | $\mathbf{24.6}_{0.3}$ | $\mathbf{13.2}_{1.6}$ | $\mathbf{19.1}_{1.6}$ | $4.6_{0.1}$ | $\mathbf{10.5}_{0.2}$ | $\mathbf{4.4}_{0.1}$ | $\mathbf{10.1}_{0.2}$ |
| $1 - TER[PL(k), PL(k')]$ | $16.0_{0.4}$ | $26.5_{0.8}$ | $17.8_{1.2}$ | $30.4_{2.3}$ | $\mathbf{4.4}_{0.1}$ | $11.1_{0.5}$ | $4.5_{0.0}$ | $10.5_{0.5}$ |

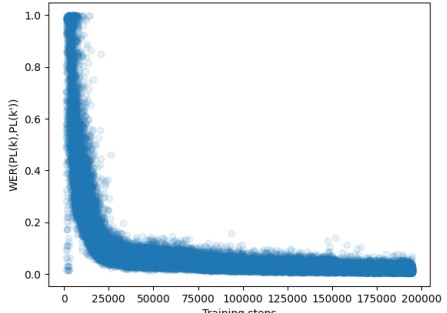
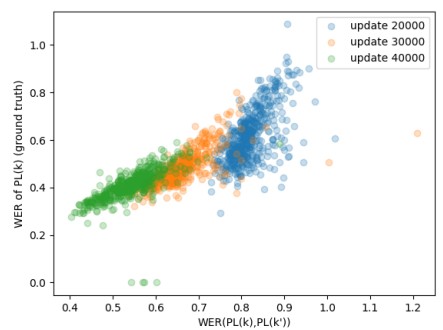

(a) $p_{out}$=WER[$PL(\boldsymbol{x}; k), PL(\boldsymbol{x}; k')$] per batch along the training.

(b) Correlation between WER[$PL(\boldsymbol{x}; k)$,golden] and WER[$PL(\boldsymbol{x}; k), PL(\boldsymbol{x}; k')$].

Figure 2: Analysis of our curriculum PLs selection criteria. WER is given in scale of (0, 1).

selection decreases WER over the baseline with constant $p_{out}$. This behavior also holds when we switch from the dynamic strategy of Eq. (3) to a constant $p_{out} = 1$ after 130K steps of training. For a 10h of labeled data setting the improvement over the baseline is larger and reaches around 1% absolute. The function $f : x \mapsto 1 - x$ performs worse than $f : x \mapsto x$ and hence we use this setting for subsequent experiments.

Our analysis of dynamic probabilities $p_{out}$ from Table 4 shows: (i) $TER[PL(\boldsymbol{x}; k), PL(\boldsymbol{x}; k')]$ is close to 100% at the beginning of training (the model changes very fast), and quickly decreases (less than 10% after 30k steps); (ii) over training different batches get different values of $p_{out}$, see Figure 2a; (iii) proposed distance correlates with the oracle WER computed between PLs and ground truth labels for $\boldsymbol{x} \in U$, see Figure 2b. The latter demonstrates that our choice of dynamic selection encapsulates knowledge about actual PLs quality.

## 5.2 ALIGNMENT SAMPLING

In Table 5 we compare results for models trained with hard PLs ($\tau = 0$), models trained with alignment sampling and constant $\tau > 0$, and models trained with a linear schedule of $\tau$ from 1 to 0.1 ($1 \to 0.1$), as described in Section 4.2. For this section we do not use dynamic control of the cache as introduced in Section 4.1. Here we highlight some observations. Firstly, alignment sampling with high $\tau$ reduces the number of diverged models (either $\tau = 1$ or $\tau = 1 \to 0.1$). Secondly, constant temperature over the training does not provide best results: $\tau = 0.1$ is similar to the baseline while $\tau = 1$ is even worse; the difference is more pronounced for the 10h of supervision with $p_{out} = 0.1 \to 1$. Besides, WER we also report TER to highlight that sampling with $\tau = 1$ leads to a notable CER degradation. However, scheduled $\tau = 1 \to 0.1$ provides both stable training (no divergence is observed in experiments) and similar or significantly better TER/WER (1.3%-2.7%) over the baseline. The best results are obtained with $p_{out} = 0.1 \to 1$ showing compatibility of sampling and dynamic probability.

Table 5: TER and WER on *dev-other* for sampling PLs with different temperature $\tau$, including linear schedule of $\tau$ in case of constant $p_{out}$ (left parts) or alternated one (right parts), see Section 3. 'DV' denotes the number of diverged models over 3 runs with random seeds. PL evolution via dynamic cache probability from Section 4.1 is not used.

| | 10h | | | | | | 100h | | | | | |
| | $p_{out} = 0.1$ | | | $p_{out} : 0.1 \rightarrow 1$ | | | $p_{out} = 0.1$ | | | $p_{out} : 0.1 \rightarrow 1$ | | |
| $\tau$ | TER | WER | #DV | TER | WER | #DV | TER | WER | #DV | TER | WER | #DV |
|---|---|---|---|---|---|---|---|---|---|---|---|---|
| 0 (argmax) | $10.1_{0.2}$ | $25.4_{0.4}$ | 0 | $7.8_{0.7}$ | $21.4_{0.3}$ | 1 | $3.9_{0.1}$ | $10.4_{0.1}$ | 1 | $3.7_{0.1}$ | $10.2_{0.1}$ | 1 |
| 0.1 | $10.9_{1.0}$ | $26.1_{2.0}$ | 0 | $8.4_{0.1}$ | $21.2_{1.9}$ | 0 | $3.9_{0.1}$ | $10.3_{0.1}$ | 1 | $\mathbf{3.6}_{0.1}$ | $10.3_{0.1}$ | 2 |
| 1 | $11.4_{1.9}$ | $26.5_{4.8}$ | 0 | $12.1_{0.6}$ | $31.2_{1.9}$ | 0 | $4.2_{0.2}$ | $10.4_{0.3}$ | 0 | $3.7_{0.1}$ | $10.4_{0.2}$ | 0 |
| $1 \rightarrow 0.1$ | $\mathbf{9.7}_{1.2}$ | $\mathbf{22.7}_{1.4}$ | 0 | $\mathbf{7.5}_{0.6}$ | $\mathbf{20.1}_{1.2}$ | 0 | $\mathbf{3.8}_{0.1}$ | $\mathbf{10.2}_{0.1}$ | 0 | $3.7_{0.1}$ | $\mathbf{10.1}_{0.1}$ | 0 |

Table 6: Combination of our methods (Sections 4.1 and 4.2) for hard labels (left part) and for sampling (right part) with a linear schedule on the temperature. 'DV' states for models divergence, 'old' denotes usage of $PL(\boldsymbol{x}; k)$, while 'new' denotes the use of $PL(\boldsymbol{x}; k')$. We compare different $p_{out}$ (all with using 'new'): scheduled $p_{out} = 0.1 \rightarrow 1$ (switching at 130K steps), $\rho = TER$ and scheduled $\rho = TER \rightarrow 1$ (switching at 130K steps). The WER on *dev-other* is reported. All results are reported across 3 runs with different seeds.

| Data | $\lambda$ | Argmax | | | | | Sampling | | | | |
| | | old | new | $0.1 \rightarrow 1$ | $\rho$ | $\rho \rightarrow 1$ | old | new | $0.1 \rightarrow 1$ | $\rho$ | $\rho \rightarrow 1$ |
|---|---|---|---|---|---|---|---|---|---|---|---|
| 10h | 1 | DV | $25.4_{0.4}$ | $21.4_{0.3}$ | $24.6_{0.3}$ | $19.1_{1.6}$ | DV | $22.7_{1.4}$ | $20.1_{1.2}$ | $21.2_{1.8}$ | $20.7_{1.9}$ |
| 10h | 5 | DV | DV | DV | DV | DV | DV | DV | DV | $14.7_{0.4}$ | $13.3_{0.2}$ |
| 100h | 1 | DV | $10.6_{0.3}$ | $11.3_{0.1}$ | $10.5_{0.2}$ | $10.1_{0.2}$ | $13.5_{0.3}$ | $10.2_{0.1}$ | $10.1_{0.1}$ | $10.5_{0.2}$ | $10.2_{0.2}$ |
| 100h | 5 | DV | DV | DV | DV | DV | DV | DV | DV | $10.7_{0.3}$ | $10.0_{0.3}$ |

## 5.3 COMBINING METHODS FOR BEST RESULTS

In this section we highlight the results that can be achieved by combining together all the methods reported above in Sections 4.1 and 4.2. In Table 6 we give a detailed comparison for both 10h and 100h of supervision. As we have now stable training pipeline from the start (no PT), we also play with a ratio $\lambda$ (see Eq. (1)) searching it in range $[1, 5]$. This raises training instability risk while larger proportion of unlabeled data may improve the model according to Likhomanenko et al. (2021a).

For 10 hours of supervised data the models benefit a lot from the higher $\lambda$ and become competitive with models trained with PT phase as well as with prior works (Baevski et al., 2020; Likhomanenko et al., 2021a)). Note that combining sampling with dynamic $p_{out}$ based on PLs evolution is necessary to have stable training for $\lambda > 1$.

To have a proper comparison with aforementioned prior works we increase the batch size and use dynamic batching for the best configuration. First, we confirm that both sampling and dynamically controlling the cache give stable training (see e.g. Appendix C Table 13). Second, in Table 7[5] for 10h/100h setup ($\lambda = 5/\lambda = 3$) our models achieve similar or better results with no PT compared to PT-based models (which are reproductions of slimIPL using the same settings that we use for our method) while matching the prior works.

To ensure our methods are general enough we probe the final configuration (found for LibriSpeech) on Common Voice, French language data. We use exactly the same models with sinusoidal positional embedding and the same hyper-parameters. The only thing we tune is slimIPL parameter $M$. Results in Table 8 show that our methods work out of the box: without PT we are able to match slimIPL baseline for 100h of supervision, while we improve results upon slimIPL for low supervision setting of 10h with an average relative WER reduction of $18\%$.

---

[5]As we use different 10h split in this work we also report results for 10h set with 24 speakers from Libri-Light used in prior works. We found that training with no PT is more prone to unstable training for this set, while our method is able to stabilize it and get comparable performance with its baseline counterpart which lags behind the prior works.

Table 7: Comparison of our best models with prior works for 10h and 100h of supervision. Results are reported across 3 random seeds. For wav2vec 2.0 and slimIPL we report the prior work results and our reproduction following official open-sourced recipes. 'Posemb' denotes type of used positional embedding. The 10h set from Libri-Light is marked with '*'.

| Model | Data | Posemb | dev WER | | test WER | |
|---|---|---|---|---|---|---|
| | | | clean | other | clean | other |
| w2v 2.0, Large (Baevski et al., 2020) | | conv | 8.1 | 12.0 | 8.0 | 12.1 |
| w2v 2.0, Large, reproduction | | conv | $8.1_{0.3}$ | $12.9_{0.2}$ | $8.1_{0.3}$ | $13.3_{0.3}$ |
| slimIPL (Likhomanenko et al., 2021a) | | relpos | 11.4 | 14.0 | 11.4 | 14.7 |
| slimIPL | 10h* | CAPE | $14.4_{0.3}$ | $18.8_{0.4}$ | $15.1_{0.4}$ | $19.3_{0.3}$ |
| Ours | | CAPE | $15.8_{1.8}$ | $20.4_{1.6}$ | $15.9_{1.5}$ | $20.4_{1.3}$ |
| slimIPL | | sinpos | $32.7_{0.6}$ | $36.8_{0.3}$ | $33.7_{0.7}$ | $37.6_{0.4}$ |
| Ours | | sinpos | $20.7_{2.0}$ | $24.4_{2.0}$ | $21.4_{2.1}$ | $24.9_{1.9}$ |
| w2v 2.0, Large | | conv | $7.4_{0.3}$ | $12.7_{0.3}$ | $7.7_{0.3}$ | $13.0_{0.4}$ |
| slimIPL | 10h | CAPE | $10.0_{0.4}$ | $15.1_{0.5}$ | $9.9_{0.4}$ | $15.7_{0.5}$ |
| Ours | | CAPE | $8.2_{0.2}$ | $13.1_{1.4}$ | $8.5_{0.2}$ | $13.6_{2.1}$ |
| slimIPL | | sinpos | $22.5_{1.3}$ | $28.1_{1.3}$ | $22.9_{1.2}$ | $29.4_{1.4}$ |
| Ours | | sinpos | $8.6_{0.2}$ | $13.3_{0.2}$ | $8.7_{0.3}$ | $13.4_{0.2}$ |
| w2v 2.0, Large (Baevski et al., 2020) | | conv | 4.6 | 9.3 | 4.7 | 9.0 |
| slimIPL (Likhomanenko et al., 2021a) | | relpos | 3.7 | 7.3 | 3.8 | 7.5 |
| slimIPL | 100h | CAPE | $3.7_{0.1}$ | $8.0_{0.1}$ | $3.9_{0.1}$ | $8.2_{0.1}$ |
| Ours | | CAPE | $4.1_{0.1}$ | $8.4_{0.1}$ | $4.0_{0.1}$ | $8.6_{0.2}$ |
| slimIPL | | sinpos | $3.7_{0.1}$ | $7.8_{0.1}$ | $3.8_{0.1}$ | $8.0_{0.1}$ |
| Ours | | sinpos | $4.0_{0.1}$ | $8.1_{0.2}$ | $4.1_{0.1}$ | $8.4_{0.2}$ |
| Lower bound, fully supervised | 960h | CAPE | $2.6_{0.1}$ | $6.9_{0.1}$ | $2.7_{0.1}$ | $6.9_{0.1}$ |

Table 8: Comparison of fully supervised, slimIPL and our methods on Common Voice French. Results are reported across 6 random seeds. Sinusoidal positional embedding is used for all models.

| Model | Data | WER | |
|---|---|---|---|
| | | valid | test |
| Fully supervised | | $59.9_{0.5}$ | $62.6_{0.6}$ |
| slimIPL | 10h | $29.9_{2.0}$ | $31.1_{2.1}$ |
| Ours | | $24.6_{1.8}$ | $26.0_{1.9}$ |
| Fully supervised | | $17.3_{0.1}$ | $19.3_{0.1}$ |
| slimIPL | 100h | $12.8_{0.2}$ | $14.1_{0.2}$ |
| Ours | | $13.0_{0.2}$ | $14.3_{0.2}$ |
| Fully supervised | 540h | $10.9_{0.4}$ | $12.3_{0.3}$ |

## 6 CONCLUSION

In this paper we show that we can perform continuous pseudo-labeling from the very start of training and get improved results in low supervision settings. We were able to achieve these results by using alignment sampling and a dynamic cache selection strategy that is based on the evolution of the pseudo-labels during training. Being able to perform pseudo-labeling from the very start further simplifies training, avoiding complicated multi-step pipelines and allows us to focus on a simpler one. Our work also provides avenues for explorations into curriculum strategies for pseudo-labeling and we hope to build upon the ideas and results presented in this paper. In the future we wish to explore the effectiveness of these methods to other settings for ASR such as sequence-to-sequence/transducer models[6], out-of-domain unsupervised data, and neural models not based on transformers.

---

[6]The proposed dynamic control of the cache does not rely on anything specific to CTC. Alignment sampling should be transferable to Transducer directly, while for sequence-to-sequence we would sample transcription directly from the model.

## 7 REPRODUCIBILITY STATEMENT

We report detailed settings of our experiments which are based on the previously open-sourced recipes for Likhomanenko et al. (2021a) through the paper and also in Appendix A.2 and B. We aim to open source the code of our method and experiments soon.

## 8 ETHICS

For this paper we used publicly available datasets. Our goal is to build models that work for low supervision settings and hope this is a positive contribution towards under-represented data sources for ASR. While one can imagine ASR being used for negative purposes, it is our hope that the advantages generated by improving ASR for low-resource settings outweigh its possible negative uses.

## ACKNOWLEDGMENTS

We would like to thank Richard He Bai, Jagrit Digani, David Grangier, Loren Lugosch, Yizhe Zhang, and machine learning research teammates for helpful discussions and support throughout the work.

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

# A    DETAILS ON EXPERIMENTAL SETUP

## A.1    SPEAKERS IN LIBRISPEECH

There is no intersection between speakers in different LibriSpeech train sets as well as in validation / test sets – all speakers are unique and are present in only one of the LibriSpeech sets. To prepare the 10h set we randomly sampled audio per speaker to gather a total 10h of audio.

## A.2    ACOUSTIC MODEL TRAINING

We keep the original 16kHz sampling rate and compute log-mel filterbanks with 80 coefficients for a 25ms sliding window, strided by 10ms which are normalized to zero mean and unit variance per input sequence before feeding into a model.

Throughout the paper we consider transformer-based models with a convolutional frontend to perform the proper striding. The encoder is composed of a 1-D convolution with kernel size 7 and stride 3 followed by 36 4-head Transformer blocks (Vaswani et al., 2017). The self-attention dimension is 768 and the feed-forward network (FFN) dimension is 3072 (with 4 heads) in each transformer block. The output of the encoder is followed by a linear layer to the output classes. We use dropout after the convolution, dropout on the self-attention and on the FFN for all transformer layers, and layer drop (Fan et al., 2020), dropping entire layers at the FFN level.

We get rid of relative positional embedding (Shaw et al., 2018) and use either sinusoidal one (Vaswani et al., 2017) or recently proposed CAPE embedding (Likhomanenko et al., 2021b) (only global shift of 30s is used): this speeds up training by 2-3x and decreases memory usage.

For SpecAugment Park et al. (2019) we follow parameters from Likhomanenko et al. (2021a): two frequency masks with frequency mask parameter $F = 30$, ten time masks with maximum time-mask ratio $p = 0.1$ and time mask parameter $T = 50$; time warping is not used.

All models are trained with CTC loss and Adagrad optimizer with linear warmup period of 64k steps, constant learning rate of 0.03 and step-wise (by 2) learning rate decay at the end of training. All models are trained on tf32 tensor cores of 8 Ampere A100 40GB GPUs for a maximum of 500k updates.

For slimIPL parameters we use always cache size of 1k. Throughout the paper we vary the proportion $\lambda$ (by default we use $\lambda = 1$ if not stated otherwise) as well as $p_{out}$. From experiments we observe that it is important to activate SpecAugment later in training (e.g. after 5k training steps) otherwise slimIPL baseline is even more prone to divergence.

## A.3    COMMON VOICE EXPERIMENTS

We use Common Voice data release from 21 July 2021[7] with French language. In total, there are 543 hours in train, 25.1h in validation and 25.8 in test sets. We randomly sample speakers from the train and take all audio belonging to the same speaker to form a 100h train subset. We end up with 982 speakers and 102h. We further sample speakers from this 100h subset to form a 10h subset: it contains 171 speakers with 11.5h. These 10h and 100h subsets are used as labeled data while the remaining 443h are used as unlabeled data. We normalize transcriptions by lower casing, removing any punctuation tokens except apostrophe, changing all diacritical marks to their corresponding English characters and removing any other non-English characters. Later, we use the same token set as for LibriSpeech.

We use the same acoustic model as for LibriSpeech experiments with sinusoidal positional embedding as all audios in Common Voice are very short (5.2s±1.5s). For fully supervised models we use dropout 0.5, 0.3 and 0.1 for 10h, 100h and 540h sets correspondingly. For slimIPL we change dropout and layer drop from 0.5 to 0.1 for 10h and from 0.3 to 0.1 for 100h, while for our methods we use dropout and layer drop of 0.1 from the beginning of training. For slimIPL we tune only parameter $M$ for the 10h setting. The rest of parameters are the same as in original slimIPL work (Likhomanenko et al., 2021a): $C$ is 1000 (100), cache probability $p_{out}$ is 0.1, data proportion $\lambda$ is 10 (3), $M$ is 40k

---

[7]https://github.com/common-voice/cv-dataset/blob/main/datasets/
cv-corpus-7.0-2021-07-21.json

(20k) for 10h (100h) setting. All models are trained with dynamic batch, same as for LibriSpeech. For our methods we use exactly the same parameters as for LibriSpeech experiments with dynamic batch.

## A.4 FULLY SUPERVISED MODELS

Table 9: Fully supervised models for 10h and 100h of LibriSpeech. Results are reported across 3 random seeds. Sinusoidal, CAPE and relative positional embeddings are denoted as 'sinpos', 'CAPE' and 'relpos' correspondingly. The 10h set from Libri-Light is marked with '*'.

| Model | Sup. set | WER | | | |
| --- | --- | --- | --- | --- | --- |
| | | dev-clean | dev-other | test-clean | test-other |
| relpos (Likhomanenko et al., 2021a) | | 31.9 | 52.3 | 32.6 | 52.4 |
| CAPE | 10h* | $37.1_{0.1}$ | $58.4_{0.1}$ | $37.7_{0.3}$ | $58.4_{0.2}$ |
| sinpos | | $76.0_{0.8}$ | $87.1_{0.5}$ | $77.1_{0.7}$ | $87.2_{0.6}$ |
| relpos | | $27.7_{0.4}$ | $48.4_{0.4}$ | $28.2_{0.3}$ | $48.8_{0.3}$ |
| CAPE | 10h | $28.2_{0.1}$ | $48.5_{0.3}$ | $28.9_{0.1}$ | $48.9_{0.2}$ |
| sinpos | | $63.4_{1.1}$ | $78.5_{0.9}$ | $64.5_{0.9}$ | $78.9_{1.1}$ |
| relpos (Likhomanenko et al., 2021a) | | 6.2 | 16.8 | 6.2 | 16.8 |
| CAPE | 100h | $5.9_{0.1}$ | $17.9_{0.1}$ | $6.2_{0.1}$ | $18.1_{0.1}$ |
| sinpos | | $6.5_{0.3}$ | $19.1_{0.2}$ | $7.1_{0.3}$ | $19.3_{0.2}$ |

## A.5 SUMMARY OF HYPER-PARAMETERS

Hyper-parameter values for both experiments on LibriSpeech and Common Voice are summarized in Tables 11, 12 and 10.

Table 10: Detailed hyper-parameters for the final experiments on Common Voice from Table 8.

| Parameter | slimIPL (10h) | Our (10h) | slimIPL (100h) | Our (100h) |
| --- | --- | --- | --- | --- |
| $M$ | 40k | 0k | 20k | 0k |
| $C$ | 1000 | 1000 | 100 | 1000 |
| $p_{out}$ | 0.1 | TER (1 after 130k) | 0.1 | TER (1 after 130k) |
| $\lambda$ | 10 | 5 | 3 | 3 |
| dropout/layer drop | 0.5→0.1 | 0.1 | 0.3→0.1 | 0.1 |
| embedding | sinpos | sinpos | sinpos | sinpos |
| $\tau$ | 0 | $\tau_k = \max(0.1, 1 - 0.1 * k/130,000)$ | 0 | $\tau_k = \max(0.1, 1 - 0.1 * k/130,000)$ |
| total batch | dynamic 290s×8 | dynamic 290s×8 | dynamic 290s×8 | dynamic 290s×8 |

Table 11: Detailed hyper-parameters for the final experiments on LibriSpeech from Table 7.

| Parameter | slimIPL (10h) | Our (10h) | slimIPL (100h) | Our (100h) |
| --- | --- | --- | --- | --- |
| $C$ | 1000 | 1000 | 100 | 1000 |
| $\lambda$ | 10 | 5 | 3 | 3 |
| dropout/layerdrop | 0.5→0.1 | 0.1 | 0.3→0.1 | 0.1 |
| $\tau$ | 0 | $\tau_k = \max(0.1, 1 - 0.1 * k/130,000)$ | 0 | $\tau_k = \max(0.1, 1 - 0.1 * k/130,000)$ |
| embedding | sinpos | sinpos | sinpos | sinpos |
| $M$ | 30k | 0k | 20k | 0k |
| $p_{out}$ | 0.1 | TER (1 after 130k) | 0.1 | TER (1 after 130k) |
| total batch | dynamic 290s×8 | 8x8 | dynamic 290s×8 | dynamic 290s×8 |
| embedding | CAPE | CAPE | CAPE | CAPE |
| $M$ | 50k | 0k | 20k | 0k |
| CAPE is used after | 0k | 25k | 0k | 5k |
| $p_{out}$ | 0.1 | TER (1 after 40k) | 0.1 | TER (1 after 130k) |
| total batch | dynamic 290s×8 | dynamic 290s×8 | dynamic 290s×8 | dynamic 290s×8 |

## B WAV2VEC AND SLIMIPL REPRODUCTION

To reproduce baselines in Table 7 for slimIPL we follow Likhomanenko et al. (2021a) and its published recipe. The only change we do is positional embedding as discussed above and batch size.

Table 12: Detailed hyper-parameters for the final experiments on LibriSpeech from Table 7 for 10h$^*$ setting.

| Parameter | slimIPL (10h$^*$) | Our (10h$^*$) |
|---|---|---|
| $C$ | 1000 | 1000 |
| $\tau$ | 0 | $\tau_k = \max(0.1, 1 - 0.1 * k/130,000)$ |
| SpecAugment | $T = 25$, 20 time masks | $T = 50$, 10 time masks |
| embedding | sinpos | sinpos |
| $M$ | 20k | 0k |
| dropout/layer drop | 0.5→0.1 | 0.5 (0.1 after 35k) |
| $\lambda$ | 10 | 1 (5 after 70k) |
| $p_{out}$ | 0.1 | TER (1 after 70k) |
| total batch | dynamic 290s×8 | 8×8 |
| embedding | CAPE | CAPE |
| $M$ | 40k | 0k |
| CAPE is used after | 0k | 70k |
| $\lambda$ | 10 | 1 (5 after 130k) |
| dropout/layerdrop | 0.5→0.1 | 0.5 (0.1 after 70k) |
| $p_{out}$ | 0.1 | TER (1 after 130k) |
| total batch | dynamic 290s×8 | 8×8 |

The rest of the training remains the same. To reproduce wav2vec 2.0 (Baevski et al., 2020) we take open-sourced Large model pre-trained on the full LibriSpeech[8] and then perform fine-tuning on our 10h set and the 10h set from Libri-Light. For fine-tuning we use open-sourced configurations for 10h[9]. We fine-tune models on 24 GPUs as specified in Baevski et al. (2020) for 3 different seeds.

## C ABLATIONS: SAMPLING FOR LARGER BATCHES

Table 13: Comparison (in WER) between different temperatures $\tau$ for sampling when large batch and longer training (600k) are used.

| $\tau$ | dev-clean | dev-other | test-clean | test-other |
|---|---|---|---|---|
| 0 (argmax) | 19.1 | 26.7 | 19.3 | 27.8 |
| $1 \to 0.1$ | 13.9 | 17.5 | 13.8 | 18.0 |

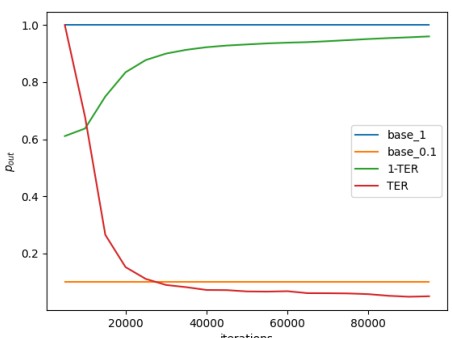
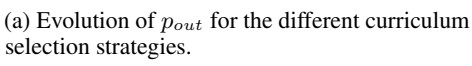

(a) Evolution of $p_{out}$ for the different curriculum selection strategies.

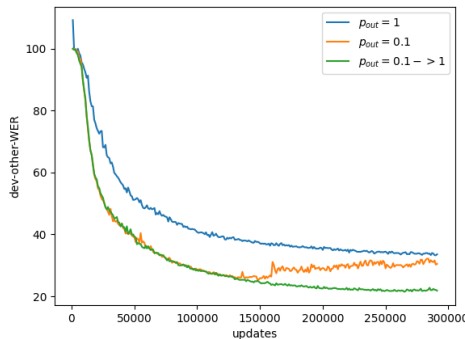

(b) Comparison between models trained with different $p_{out}$: constant 1 (blue) or 0.1 (orange), or scheduled $0.1 \to 1$ (green).

Figure 3: Analysis of the probability $p_{out}$.

---

[8]Released at `https://dl.fbaipublicfiles.com/fairseq/wav2vec/libri960_big.pt`.
[9]They are availble at `https://github.com/facebookresearch/fairseq/blob/main/examples/wav2vec/config/finetuning/vox_10h.yaml`.

