# OpenReview forum: "Continuous pseudo-labeling from the start"
_ICLR.cc/2023/Conference — ICLR 2023 poster_

### Official Review · Reviewer_ZfEP · 2022-10-23

**Confidence:** 3
**Clarity, Quality, Novelty And Reproducibility:** 1) The paper is clear overall for a r…
**Correctness:** 3
**Technical Novelty And Significance:** 3
**Empirical Novelty And Significance:** 3
**Recommendation:** 6

**Strength And Weaknesses:**

Strengths:

1) The overall presentation and flow of ideas is good. The paper does a good job in setting up the motivation for the work by presenting some early results.
2) Extensive ablation studies over each component of the proposed methodology helps the paper.
3) Results are compared with comprehensive baselines.
4) Details about the model architecture and other results included for reproducibility.

Weaknesses:

1) The major part of this paper builds over the previously proposed slimIPL. Yet, not enough explanation is given about slimIPL in the main text. This might be inconvenient for a reader not conversant in the algorithm. Having a full algorithm in the main text will help the paper.
2) The algorithm for slimIPL (appendix A) confused me. In the 3rd step, when the cache is being initialized, why is there an update step using (x,y) in L? Hasn’t that already been done prior to generating the cache? Or is this a typo?
3) It is not very clear why alignment sampling should work. The authors say that it provides effective stabilization by adding noise to the targets. But the targets are noisy anyway during the start of training as the TER is close to 100% initially. So why would sampling produce targets that are any different? There is not enough explanation for this.
4) All results are shown on CTC based models. Do these techniques translate to encoder-decoder based ASR models like LAS and RNN-T which are very popular? The authors do not comment on this.
5) In the final results, compared to w2v 2.0 results, the proposed techniques work better only on dev-other and test-other but not the clean subsets using 10h of speech. Why should the proposed method be chosen over w2v2.0 ?
6) Minor issues: The caption for figure 1 can be more informative. What does “CAPE” and “sinpos” mean in table 7?


**Summary Of The Paper:**

This paper proposes novel changes to an existing self training algorithm for ASR such that performance is improved without a pretraining step. Specifically, the proposed model generates pseudo labels (PL) for unlabelled data from the very start of supervised training and uses it to augment the dataset. This is done by dynamically evolving a cache of data by using a simple yet effective filtering criteria for unlabelled examples. Furthermore, to avoid instability in the proposed PL generation from the start an alignment sampling is used which samples tokens from the predicted distribution instead of an argmax. Results show improvement over previous work.


**Summary Of The Review:**

This paper contributes to the field of self training for ASR models. The overall presentation of ideas is good and the proposed techniques are simple yet effective. The authors conduct extensive ablation studies to back their claims. However, the paper could benefit by explaining some intuition behind the alignment sampling criteria. The fact that the authors do not comment on non-CTC based models and that w2v2.0 still works better on 2 out of 4 test sets, does weaken the paper.

---

> ### Author Response · Authors · 2022-11-11
> **Authors' response**
>
> Dear Reviewer ZfEP,
>
> Thanks for your time and detailed feedback. Please find below our responses to your comments.
>
> - *"The major part of this paper builds over the previously proposed slimIPL. Yet, not enough explanation is given about slimIPL in the main text. This might be inconvenient for a reader not conversant in the algorithm. Having a full algorithm in the main text will help the paper."*  We chose to describe the slimIPL algorithm in the Appendix, to highlight only our contributions (alignment sampling and new curriculum selection of pseudo-labels). Unfortunately, this emphasis seems to have made the contrast with slimIPL harder to follow. We will attempt to improve this by providing a clearer explanation of our modifications to slimIPL and by highlighting the differences.
> - *"The algorithm for slimIPL (appendix A) confused me. In the 3rd step, when the cache is being initialized, why is there an update step using (x,y) in L? Hasn’t that already been done prior to generating the cache? Or is this a typo?"* This is not a typo, and according to the original slimIPL paper this choice improves the stability of the continuous training procedure. One way to understand the importance of this is to realize that this fills the cache with samples that encapsulate the model state over multiple different time steps. This gives us a cache where the predictions come from an ensemble of models – which can make the training more stable, reminiscent of replay buffers in Deep Reinforcement learning.
> - *"It is not very clear why alignment sampling should work. The authors say that it provides effective stabilization by adding noise to the targets. But the targets are noisy anyway during the start of training as the TER is close to 100% initially. So why would sampling produce targets that are any different? There is not enough explanation for this."* Alignment sampling is a way to enforce a lower bound on the entropy of the label distribution which mitigates the collapse issue. With no regularization (cache, and/or alignment sampling), the PL procedure often collapses to generating just blanks very quickly – it is biased, has 100% TER, as you suggested, but has no variance. Alignment sampling avoids this by generating noisy targets that have variance.
> - *"All results are shown on CTC based models. Do these techniques translate to encoder-decoder based ASR models like LAS and RNN-T which are very popular? The authors do not comment on this."* The proposed dynamic control of the cache does not rely on anything specific to CTC: we compute the TER between "old" and "new" PLs. The alignment sampling should be transferable to Transducer and seq2seq models too – Transducers are an extension of CTC and the same method should be applicable, while for Seq2Seq models we would sample transcriptions from the model (Please see our response to Reviewer bgqJ on a similar question).
> - *"In the final results, compared to w2v 2.0 results, the proposed techniques work better only on dev-other and test-other but not the clean subsets using 10h of speech."* It is more challenging to improve on dev-other/test-other, and thus hyper-parameters tuning was performed according to the dev-other set.
> - *"Why should the proposed method be chosen over w2v2.0 ?"* wav2vec 2.0 and pseudo-labeling were shown to be complementary (e.g. https://arxiv.org/pdf/2010.11430.pdf ICASSP 2021), so both directions are worth investigating. In addition, pseudo-labeling implementation and hyper-parameter tuning are easier than wav2vec 2.0 which require pretraining with a different objective. PL also requires less training resources.
> - *"Minor issues: The caption for figure 1 can be more informative. What does “CAPE” and “sinpos” mean in table 7?"* We will make the caption for Figure 1 more informative. For table 7, sinpos refers to the classical sinusoidal positional embeddings used for the transformers whereas CAPE refers to the recently proposed positional embedding augmentation strategy (https://arxiv.org/pdf/2106.03143.pdf NeurIPS 2021).
>
> Let us know if you have any other questions or suggestions on how to improve the work!
>
> Best regards,
> Authors.

---

> > ### Comment · Reviewer_ZfEP · 2022-11-22
> > **Further comments**
> >
> > I thank the authors for their response. Most of my questions were answered. Although I still think there is no empirical evidence given by the authors to show that the proposed techniques will transfer to encoder-decoder based methods. But this is not a huge issue.
> >
> > I would keep my score to 6 given that most of this paper is moderately novel and largely builds on previous work.

---

### Official Review · Reviewer_bgqJ · 2022-10-23

**Confidence:** 4
**Correctness:** 3
**Technical Novelty And Significance:** 2
**Empirical Novelty And Significance:** 3
**Recommendation:** 6

**Clarity, Quality, Novelty And Reproducibility:**

As mentioned in the strength and weakness.
I believe the reproducibility is good.

**Strength And Weaknesses:**

Strength:
1. Analyzed problems of previous methods (training unstable, easy to diverge) and proposed corresponding solutions (PL evolution and alignment sampling). The solutions are proved to be effective through experiments on Librispeech dataset.

Weakness
1. The writing style is kind of confusing: could be better at explaining the main process of the proposed method and the difference with and improvements over slimIPL.
2. The results in table 5 and 6 use different training hyperparameters compared to table 7. Not sure why this inconsistency exist but it could be clearer comparison under the same exp settings.
3. This line of research (PL for ASR) all uses Librispeech for evaluation. I think this limits the potential value for real scenario. The most suitable scenario of this technique is where we have some labeled data to train an initial usable model, then we utilize unlabeled data collected before hand or even periodically collect unlabeled data from the product, to which the ASR system is applied. In this case, the problem could be much more complex, e.g. the distribution of the initial labeled data could be different from the unlabeled data collected later, or even the unlabeled data have different distribution at different time steps. The conclusion drew from this paper (or other PL related studies) could not be suitable for this scenario. I believe if the authors add exps on more real data, the conclusion could be more convincing and thus more contribution to the community (like studies in https://www.isca-speech.org/archive/interspeech_2022/baby22_interspeech.html, https://arxiv.org/abs/2207.09078)
4. The alignment sampling methods assume a CTC decoder, which may not be applicable to RNN-T and LAS ASR models.

**Summary Of The Paper:**

This paper extends previous work on pseudo labeling for ASR (slimIPL to be specific) and achieve similar performance while much stable training process without even using the initial supervised training.
Generally, pseudo-labeling has been proved to be very valuable to industry applications of ASR because usually industries have labeled data for bootsrapping, and also have huge amount of unlabelled data for generation of pseudo labels. So the work in this study could be of great value to real applications.

**Summary Of The Review:**

Overall, I recognize the value of value of this paper as further improvement in the research about PL for ASR, though marginal. I hope more (realistic) evaluations could be done to bring more benefits to the community.

---

> ### Author Response · Authors · 2022-11-11
> **Authors' response**
>
> Dear Reviewer bgqJ,
>
> Thanks for your time and detailed feedback. Please find below our responses to your comments.
> - *"The writing style is kind of confusing: could be better at explaining the main process of the proposed method and the difference with and improvements over slimIPL."* Thank you for pointing this out. We chose to describe the slimIPL algorithm in the Appendix, and focus the paper on our contributions alone (alignment sampling and new curriculum selection of pseudo-labels). Unfortunately, this emphasis seems to have made the contrast with slimIPL harder to follow. We will attempt to improve this by providing a clearer explanation of our modifications to slimIPL and by highlighting the differences.
> - *"The results in table 5 and 6 use different training hyperparameters compared to table 7. Not sure why this inconsistency exist but it could be clearer comparison under the same exp settings."* Tables 5 and 6 represent exploratory experiments in which we investigated various ablations. To limit the compute required, these experiments were run with a smaller grid of hyper-parameters. Once we had the best settings from these ablations, we widened the grid of hyper-parameters to achieve the best results (in Table 7).
> - *"This line of research (PL for ASR) all uses Librispeech for evaluation. I think this limits the potential value for real scenario. The most suitable scenario of this technique is where we have some labeled data to train an initial usable model, then we utilize unlabeled data collected before hand or even periodically collect unlabeled data from the product, to which the ASR system is applied. In this case, the problem could be much more complex, e.g. the distribution of the initial labeled data could be different from the unlabeled data collected later, or even the unlabeled data have different distribution at different time steps. The conclusion drew from this paper (or other PL related studies) could not be suitable for this scenario. I believe if the authors add exps on more real data, the conclusion could be more convincing and thus more contribution to the community (like studies in https://www.isca-speech.org/archive/interspeech_2022/baby22_interspeech.html, https://arxiv.org/abs/2207.09078)."* We focused primarily on LibriSpeech because this is the dataset on which the community has well established benchmarks for self-supervised and semi-supervised learning. Training without pre-training phase is challenging even for in-domain data, so we started there, and hope to demonstrate its feasibility on out-of-domain scenarios for future work. Note that this setup does have some elements of distributional shift in audio quality – in both the 10h and 100h experiments the labeled data comes from the clean subset of LibriSpeech, while the unlabeled data comes from a mix of the remaining clean, and noisy subsets of LibriSpeech.
> - *"The alignment sampling methods assume a CTC decoder, which may not be applicable to RNN-T and LAS ASR models."* The idea should be transferable to Transducer and seq2seq models too – Transducers can be viewed as an extension of CTC (one could sample from the encoder output), while for Seq2Seq models we would sample transcriptions from the model. Although not directly related to our method, we also refer the reviewer to https://arxiv.org/abs/1609.00150, https://arxiv.org/abs/1506.03099 and https://arxiv.org/pdf/1511.06732.pdf, papers that show how sampling transcriptions in Seq2Seq models can be beneficial.
>
> Let us know if you have any other questions or suggestions on how to improve the work!
>
> Best regards,
> Authors.

---

### Official Review · Reviewer_fsR2 · 2022-10-24

**Confidence:** 3
**Correctness:** 4
**Technical Novelty And Significance:** 3
**Empirical Novelty And Significance:** 3
**Recommendation:** 5

**Clarity, Quality, Novelty And Reproducibility:**

The paper is clear, easy to follow and to understand. The proposed method is well described and illustrated. The evaluation is good but limited to one dataset only.

The analysis of the evolution of the Levenshtein distance between successive pseudo-label generation is interesting, as is the further analysis in the appendix.

The proposed method is mostly improvements of existing techniques, but they are relevant and address a relevant problem, the choices are well motivated and seem novel enough.

The paper looks clear enough for reproduction, and the authors state that the code will be made available.

Minor question:
  Do tables 5.1 and 5.2 contain results using only one of the methods (only cache strategy / only alignment sampling)? I think this could be clearer in the text.

**Strength And Weaknesses:**

Strength: The idea is interesting and the results are good. Self-training from the start is indeed useful. The proposed improvements of slimIPL are relevant and well motivated and analysed.

Weaknesses: The proposed method is only evaluated on setups simulated from LibriSpeech. There is no comparison with training with all the labeled data of LibriSpeech. Moreover, the 10h scenario where the proposed approach beats other methods does not look like the one usually explored for this problem: this could be motivated more in the text. The approach described in this paper is not as good as prior work on usual 10h used.  It would also have been interesting to see some results for actual low-resource scenarios.

**Summary Of The Paper:**

This paper presents a method to self-train ASR systems directly, without requiring an initial pre-training stage, which is especially useful in low-resource setups. Similarly to an existing method, slimIPL, the proposed approach generates pseudo-labels as training progresses and maintains a cache of pseudo-labels, regularly updated. The main differences are that the proposed method does not require a first pre-training step, that the pseudo-labels are updated when they are used before putting them back in the cache, and that the probability of removing a pseudo-label from the cache is computed from the difference between the older and updated labels. Additionally, the pseudo-labels are sampled from the predicted token distributions, whereas slimIPL generates 1-best labels.

The method is evaluated on LibriSpeech with two simulated setups corresponding to 10 and 100h hours of labeled data. The impact of the different hyper-parameters and proposed techniques and tricks are evaluated and the results show a more stable training and improved word error rates.

**Summary Of The Review:**

The paper is quite good, nice to read and well explained. The motivation for this idea is interesting and solves a real issue. I think this is worth sharing with the community, but I would like to see more results for different scenarios than just LibriSpeech.

---

> ### Author Response · Authors · 2022-11-11
> **Authors' response**
>
> Dear Reviewer fsR2,
>
> Thanks for your detailed feedback. Please find our responses to your suggestions and questions below:
> - *"Minor question: Do tables 5.1 and 5.2 contain results using only one of the methods (only cache strategy / only alignment sampling)? I think this could be clearer in the text."* Indeed, only one of the methods was used here because we wanted to show the disentangled results first. In section 5.3, "Combining methods for best results", we present results when both strategies are in use. We will clarify this in the next version.
> - *"The proposed method is only evaluated on setups simulated from LibriSpeech; The evaluation is good but limited to one dataset only; I would like to see more results for different scenarios than just LibriSpeech."* We focused primarily on LibriSpeech because this is the dataset on which the community has well established benchmarks for self-supervised and semi-supervised learning. We will do our best to provide results on another dataset in time for the rebuttal.
> - *"There is no comparison with training with all the labeled data of LibriSpeech."* Fully supervised training with the same architecture trained on full LibriSpeech (960h) as labeled data gives the following results: **dev-clean 2.6 ± 0.1, dev-other  6.9 ± 0.1, test-clean  2.7 ± 0.1, test-other  6.9 ± 0.1**. As you can see from our experiments, pseudo-labeling almost matches these numbers using labels for only 100h on the clean set, instead of the labels for all 960h. We will add these numbers.
> - *"Moreover, the 10h scenario where the proposed approach beats other methods does not look like the one usually explored for this problem: this could be motivated more in the text."* Prior works reconstructed the 10h set by selecting it from both clean and noisy subsets of LibriSpeech. The 100h LibriSpeech set on the other hand is constructed only from clean data. To keep our experiments consistent, and also to assess domain transfer to the unlabeled noisy subsets, we reconstructed the 10h set from the 100h clean subset of LibriSpeech, sampling randomly from the speakers.
> - *"The approach described in this paper is not as good as prior work on usual 10h used."* For faster experimentation we used the simplest of positional embedding schemes (sinusoidal embedding) in this paper, as opposed to relative positional embedding which is more compute intensive but produces better results. We will run ablations with other positional embedding schemes and will provide results in the review cycle.
> - *"It would also have been interesting to see some results for actual low-resource scenarios."* There is a history of works comparing results only on LibriSpeech, which motivated our dataset choices. We are working on additional experiments with the Common Voice dataset.
>
> Let us know if you have any other questions or suggestions on how to improve the work!
>
> Best regards,
> Authors.

---

### Official Review · Reviewer_6PJR · 2022-10-27

**Confidence:** 4
**Correctness:** 4
**Technical Novelty And Significance:** 3
**Empirical Novelty And Significance:** 4
**Recommendation:** 8

**Clarity, Quality, Novelty And Reproducibility:**

**Clarity**
Extremely clear.

**Quality**
Very high.

**Novelty**
OK.

**Reproducibility**
Should be reproducible.

**Strength And Weaknesses:**

**Strength**
- The experiment design is very well motivated and executed.
- The resulting method is both simple in recipe (1 stage training), also achieve significant WER reduction on 10h dataset.
- The paper writing is extremely clear.

**Wakenesses**
- Limited novelty.
- Absence of hyperparameter table in the appendix.

**Summary Of The Paper:**

**Summary**
This paper considers continuous self-training for ASR. This paper build on the observation that a previous method slimIPL performance degrades as the number of pretraining step M increase. The authors hypothesize that pretraining would cause overfitting when the supervised data is limited. To address this mentioned concerns, the authors propose self-training from the beginning of process. The authors introduces a series of tricks to improve the robustness and convergence of slimIPL.
- regenerating PL when returning a sample to cache
- dynamically compute the returning probability with Levenshtein edit-distance
- sampling PL (with temperature scheduling) instead of using 1-best

The optimal recipe achieve similar result with the original paper on 100h dataset but much better on 10h dataset.

**Summary Of The Review:**

accept

---

> ### Author Response · Authors · 2022-11-09
> **Authors' response**
>
> We would like to thank Reviewer 6PJR for the detailed and supportive feedback. We address the weaknesses raised below:
> - *"Limited novelty"*: We agree that our research is built on top of existing algorithms, and uses some prior methods (e.g. sampling, although prior work did not apply a temperature schedule). Our main contribution is to show that we can perform pseudo-labeling from the start with no supervised pre-training. We achieve this using a new curriculum to select pseudo-labels, conditioned on the pseudo-labels’ evolution (tracked via the edit distance between “old” and “new” pseudo-labels).
> - *"Absence of hyperparameter table in the appendix."* We tried to cover all the details in the Appendix B, but will happily provide a tabular summary in the upcoming revision.
>
> Let us know if you have any other questions or suggestions on how to improve the work!
>
> Best regards,
> Authors.

---

### Author Response · Authors · 2022-11-18
**Additional experiment on Common Voice data and paper revision**

Dear Reviewers,

Several of you raised the question about the applicability to other datasets than LibriSpeech. We ran an additional experiment to show the applicability of our method on the French language on the Common Voice dataset, which has 540 hours of labeled data. We performed stratified sampling by sampling speakers first and then sampling utterances from the speakers to form labeled 10h and 100h subsets; the remaining 440h was used as the unlabeled data (there is no speaker overlap between labeled and unlabeled data). We normalized transcriptions by lower casing, removing any punctuation tokens except apostrophe, changing all diacritical marks to their corresponding English characters and removing any other non-English tokens.


We used exactly the same model and hyper-parameters as in Table 7. The results we got are:

|   Model    |  Sup. set     |   valid    | test      |
|---    |---    |---    |---    |
| Fully supervised | 10h |  59.9 +/- 0.5 | 62.6 +/- 0.6 |
| slimIPL | 10h | 29.9 +/- 2.0 | 31.1 +/- 2.1 |
| Ours | 10h | 24.6 +/- 1.8 |  26.0 +/- 1.9 |
| Fully supervised | 100h | 17.3 +/- 0.1 |19.3 +/- 0.1 |
| slimIPL | 100h | 12.8 +/- 0.2 | 14.1 +/- 0.2 |
| Ours | 100h | 13.0 +/- 0.2 | 14.3 +/- 0.2 |
| Fully supervised | 540h | 10.9 +/- 0.4 | 12.3 +/- 0.3 |

Results show that our methods work out of the box: without pre-training phase on labeled data only we are able to match slimIPL baseline for 100h of supervision, while we improve results upon slimIPL for low supervision setting of 10h with an average relative WER reduction of 18%.

We have incorporated all comments and updated Table 7 with few models finished in time. All changes are marked with red color for readability.

Best regards,
Authors.

---

### Decision · Program_Chairs · 2023-01-20

**Decision:**

Accept: poster

**Justification For Why Not Higher Score:**

1. Similar ideas have been proposed [1]. Thus, the proposed algorithm is not completely new.

[1] Likhomanenko, Tatiana, et al. "slimipl: Language-model-free iterative pseudo-labeling." arXiv preprint arXiv:2010.11524 (2020).

**Justification For Why Not Lower Score:**

1. This paper presents a continuous self-training method that can train an ASR model from the start without any supervised pre-training. The experimental results have confirmed the effectiveness of the proposed approach.
2. The present work has analyzed problems of previous methods (training unstable, easy to diverge) and proposed corresponding. The work in this study could be of great value to real applications.

**Metareview: Summary, Strengths And Weaknesses:**

1. This paper presents a continuous self-training method that can train an ASR model from the start without any supervised pre-training. The experimental results have confirmed the effectiveness of the proposed approach.
2. The present work has analyzed problems of previous methods (training unstable, easy to diverge) and proposed corresponding. The work in this study could be of great value to real applications.
3. Similar ideas have been proposed [1]. Thus, the proposed algorithm is not completely new.

[1] Likhomanenko, Tatiana, et al. "slimipl: Language-model-free iterative pseudo-labeling." arXiv preprint arXiv:2010.11524 (2020).

**Note From Pc:**

if the above contains the word "oral" or "spotlight" please see: "oral" presentation means -> notable-top-5% and "spotlight" means -> notable-top-25%. As stated in our emails, we are disassociating presentation type from AC recommendations

**Summary Of Ac-Reviewer Meeting:**

The scores from the reviewers are quite consistent, and thus there was no AC-reviewer meeting for this paper.